# Efficacy of a Single Injection of Stromal Vascular Fraction in Dogs with Elbow Osteoarthritis: A Clinical Prospective Study

**DOI:** 10.3390/ani14192803

**Published:** 2024-09-28

**Authors:** Yvonne Bruns, Maike Schroers, Stephanie Steigmeier-Raith, Anja-Christina Waselau, Sven Reese, Andrea Meyer-Lindenberg

**Affiliations:** 1Clinic of Small Animal Surgery and Reproduction, Ludwig-Maximilians-Universität München, 80539 Munich, Germany; bruns.yv@gmail.com (Y.B.); s.steigmeierraith@lmu.de (S.S.-R.); a.waselau@lmu.de (A.-C.W.); ameylin@lmu.de (A.M.-L.); 2Department of Veterinary Sciences, Ludwig-Maximilians-Universität München, 80539 Munich, Germany; sven.reese@lmu.de

**Keywords:** osteoarthritis, cubarthrosis, regenerative cells, stromal vascular fraction, dog

## Abstract

**Simple Summary:**

Osteoarthritis is a degenerative joint disease. Chronic joint inflammation leads to a loss of joint cartilage. Osteoarthritis is the most common joint disease in dogs and usually develops secondarily as a result of joint incongruence or trauma. Dogs with advanced osteoarthritis often suffer from chronic pain, reduced quality of life and restricted mobility despite pain medication. Conventional treatment options for OA can alleviate the symptoms, but cartilage regeneration is not achieved. Regenerative medicine is increasingly becoming the focus of osteoarthritis research because regenerative cells contribute to tissue healing. The aim of this study was to measure the treatment success of regenerative cell therapy in dogs with chronic osteoarthritis that were no longer responding to pain therapy. The regenerative cells were obtained from the dogs’ fatty tissue, processed and applied to the affected joints. Based on the veterinary lameness examination, gait analysis and owner questionnaires, it was assessed whether the regenerative cell therapy led to an improvement in the dogs’ lameness and pain. In addition, the animals were X-rayed to measure the progression of osteoarthritis. Some patients showed a therapy-related improvement in lameness and pain. However, the therapy did not have an effect on all patients.

**Abstract:**

This study aimed to assess the efficacy of a single intra-articular injection of autologous stromal vascular fraction (SVF) in dogs with chronic lameness due to advanced elbow osteoarthritis (OA) that were unresponsive to conventional drug therapy. In this clinical, prospective, non-blinded, single-center study, twenty-three dogs received autologous SVF derived from falciform adipose tissue. Primary outcome measures over the six-month study period included clinical-orthopedic and radiographic examinations, objective gait analysis and validated owner questionnaires. In 19 of 23 joints, no progression of OA was visible radiographically. Peak vertical force improved significantly at three months and vertical impulse at six months after the injection compared to baseline. Over 33% of dogs demonstrated treatment-related improvements in lameness based on objective gait analysis. Owner questionnaires indicated significant improvement in clinical signs throughout the study period and 26% of dogs showed treatment-related improvements in pain scores according to the Canine Brief Pain Inventory. No side effects were reported. These findings suggest that autologous regenerative cell therapy may provide a promising treatment option for dogs with advanced OA that do not respond to conventional drug therapy. However, the treatment did not improve the clinical symptoms in all dogs, so it cannot be recommended for all patients.

## 1. Introduction

Osteoarthritis (OA) is a degenerative joint disease that can be extremely painful and is estimated to affect 15% of dogs in Germany [1]. In North America, approx. 20% of young dogs and approx. 80% of geriatric dogs (>8 years old) develop OA [2]. There are many causes of OA of large joints, but developmental diseases such as dysplasia of the hip or elbow joints are often the underlying cause of OA [3]. Elbow dysplasia (ED) is the most common disease of the elbow joint in dogs [4] and despite various surgical treatment options, the development of cubarthrosis cannot be prevented [5,6]. If there is an incongruence in the elbow joint, joint debridement and the removal of loose cartilage fragments are usually not sufficient to prevent the development of medial compartment syndrome [7]. OA leads to degeneration of the articular cartilage, subchondral bone sclerosis, osteophytosis, synovitis and degeneration of ligaments [8]. Dogs with chronic OA often suffer from pain and limited mobility despite treatment with medication, resulting in a reduced quality of life [9,10]. 

They are often treated with non-steroidal anti-inflammatory drugs (NSAIDs) [4,11]; however, a large number of side effects of the common NSAIDs are known [12]. Knowledge of the side effects of NSAIDs is widespread among veterinarians and owners and leads to a certain caution in giving NSAIDs in the long term. According to Belshaw et al. (2016), this could lead to inadequate analgesia in dogs with OA [13]. Therefore, it seems important to use other analgesic therapeutics if necessary. There are other pharmaceutical treatment options, but these showed a weaker analgesic effect than NSAIDs, were sometimes insufficient in dogs with OA or currently lack sufficient evidence [11,14,15,16,17]. Monoclonal antibodies against nerve growth factor are also increasingly being used, and although they reduce the OA-associated pain [18], they cannot repair cartilage damage itself.

Mesenchymal stem cells (MSCs) have become the focus of OA therapies [19]. They can differentiate into various mesodermal cell types [20]. Although MSCs may differentiate into chondrocytes, they appear to have a beneficial effect in OA due to their immunomodulatory paracrine capacity rather than in replacing cartilage damage [21]. In vivo, MSCs migrate to the site of inflammation (so-called homing) and release bioactive factors [22]. They have a trophic, immunomodulatory and anti-inflammatory effect at the site of inflammation [23,24,25].

An experimental study demonstrated a reduced inflammatory response in a canine cruciate ligament rupture model after intra-articular injection of allogeneic adMSCs compared to the control group by measuring TNF-α, COX-2, IL-1β, iNOS and IFN-γ [26]. After intravenous and intra-articular injection of autologous bone marrow mesenchymal stem cells (BM-MSCs), reduced systemic and joint inflammation, including reduced IFN-γ and CrP concentration, was measured [27]. In addition, experimental studies in dogs have shown that MSC improves cartilage regeneration according to histomorphological criteria [26,28,29]. The efficacy of MSCs in the treatment of OA in dogs has also been demonstrated after clinical application [30,31,32]. However, treatment of canine elbow OA with regenerative cells has shown contradictory results in previous studies, as measured by owner surveys and gait analysis [33,34,35].

As there are no consistent results in the available literature regarding the efficacy of MSC in dogs with elbow OA, the aim of this prospective clinical study was to evaluate regenerative cell therapy in dogs with advanced OA of the elbows over a period of six months. All dogs showed chronic lameness despite previous surgical and medical treatment of medial compartment disease. A modern and time-saving point-of-care procedure (ARC TM system, InGeneron GmbH, Houston, TX, USA) was used to obtain the SVF, which was established in the in-house laboratory.

## 2. Materials and Methods

### 2.1. Study Design

This study was conducted as a clinical, prospective, non-randomized, non-blinded, single-center study at the Clinic for Small Animal Surgery and Reproduction, LMU Munich, and was approved by the Ethics Committee of the Centre of Faculty of Veterinary Medicine of the LMU (No. 31-20-06-2014).

Dogs with chronic lameness due to previously diagnosed and surgically treated medial compartment disease of the elbow joint were included in the study. To be included, the dogs had to exhibit lameness, for at least three months, that no longer showed any improvement despite adequate pain therapy. A specific treatment protocol was not mandatory prior to inclusion in the therapy, as the choice of painkiller was based on the dogs’ tolerance. There was no requirement for the dogs to have received nutraceuticals or physiotherapy prior to inclusion in the study; the dogs receiving these treatments prior to inclusion had to continue them constantly during the study in order to rule out any influence on the study results. Study patients who underwent bilateral arthroscopy for medial compartment disease and had bilateral OA with clinical relevance were included in the study for bilateral treatment with SVF. Unilaterally operated dogs that had only unilateral clinically relevant OA were treated unilaterally with SVF. Elbow joint surgery had to have been performed at least six months previously and joint injections in the elbow joint had to have been performed at least three months previously.

Further inclusion criteria were a body weight between 15 kg and a maximum of 60 kg, and a minimum age of nine months. Prerequisites for participation in the study were good general health and unchanged laboratory parameters. They had to show clear lameness (visual lameness score grade two or higher) on one limb, as well as a positive pain response and pathological findings such as crepitation or joint filling. As part of the preliminary examination, the affected elbow joint and that of the contralateral side were examined orthopedically and radiographically. A modified OA score according to the protocol of the International Elbow Working Group (IEWG score) of one was a prerequisite for inclusion in the study. During the gait analysis examination, the dogs had to be cooperative and show clear, measurable lameness on one limb, even if they were affected on both sides. Dogs that were uncooperative on the treadmill were excluded. Exclusion from the study was also due to unclear or unmeasurable lameness of the forelimb during the orthopedic or gait analysis examination. Other orthopedic diseases of the affected limb, of other limbs or of the spine with clinical relevance also led to exclusion from the study. If bone fragments were found radiologically in the affected joint despite a previous CT scan, the dogs were also excluded from the study. If the study patients developed other orthopedic diseases during the study period, they were excluded from further follow-up examinations.

The owners had to agree to return to the follow-up examinations for up to six months after the injection of the regenerative cells. Early termination of the study was possible after examination of the dog for three months if the owners wanted to have another therapy carried out for the dog and the regenerative cell therapy had led to a worsening of or no improvement in lameness and pain. The owners had to agree not to start any other therapy during the study period and to discuss changes in medication with the study leaders. No patient received permanent analgesic therapy during the study period; emergency administration of an NSAID or metamizol was permitted in the event of worsening lameness or clear signs of pain. This had to be discontinued at least 24 h before all examinations.

### 2.2. Cell Collection, Preparation and Injection

This study used the “Transpose RT cell therapy” from InGeneron (InGeneron GmbH, Munich, Germany), which can be used to obtain stromal vascular fraction (SVF) from the adipose tissue of dogs for autologous regenerative cell therapy. The falciform fat tissue was routinely removed under intubation anesthesia in the operating room of the Clinic for Small Animal Surgery and Reproduction, LMU Munich, using diazepam (0.5 mg/kg i.v.) (Solupam^®^, Dechra Veterinary Products Deutschland GmbH, Aulendorf, Germany), Propofol (2–4 mg/kg i.v.) (Narcofol, CP-Pharma Handelsgesellschaft mbH, Burgdorf, Germany), Isoflurane (1.2–1.4 volume%) (Isoflurane CP^®^, Fa. CP-Pharma Handelsgesellschaft mbH, Burgdorf, Germany) and, for analgesia, methadone (0.2 mg/kg i.v.) (Comfortan^®^, Dechra Veterinary Products Deutschland GmbH, Aulendorf, Germany). During the induction of anesthesia, the dogs received a single intravenous dose of amoxicillin—clavulanic acid (12.5 mg/kg i.v.) (AmoxiClav, CP-Pharma Handelsgesellschaft mbH, Burgdorf, Germany). The adipose tissue (approx. 10 g) was removed via an incision (3–5 cm) in the linea alba. It was immediately transported to the in-house laboratory in a sterile centrifuge tube for further processing. After a routine closure of the laparotomy wound, the patients received a single dose of buprenorphine (0.02 mg/kg i.v.) (Buprenodale, Dechra Veterinary Products Deutschland GmbH, Aulendorf, Germany). The fatty tissue was sterilely minced in the in-house laboratory and processed in a sterile centrifuge tube according to the manufacturer’s instructions (InGeneron GmbH, München, Deutschland). The production of the SVF has already been described in more detail by the authors [36,37]. The SVF was mixed with 1 mL of Ringer’s lactate per joint to be treated. A quantity of 0.1 mL of the prepared solution was isolated using a live/dead stain and tested for cell count and viability (Trypan blue and SYTO™ 13, Invitrogen, (Waltham, MA, USA)). A quantity of 0.9 mL of the solution was injected into the affected elbow joint from the medial side under sterile conditions immediately after preparation [37]. 

The dogs were discharged on the same day and the owners were given an NSAID as required and amoxicillin—clavulanic acid (Amoxiclav, CP-Pharma Handelsgesellschaft mbH, Burgdorf, Germany) (12.5 mg/kg two times a day) for a further 5 days. The stitches were removed after 10–14 days.

### 2.3. Outcome Measures

During the orthopedic examination, the elbow joints were palpated to detect any increased joint filling. The joints were passively manipulated so that restricted mobility as well as painful defensive reactions or crepitation could be recorded.

To determine the visual lameness score, the patients were presented by the owners walking and trotting on a lead. The degree of lameness was assessed by the same two veterinarians (first and second author) according to the following scheme: Grade 0: no lameness; Grade 1: indistinct low grade; Grade 2: low grade; Grade 3: moderate grade (the limb is not loaded intermittently); and Grade 4: high grade. A clinical improvement in lameness was defined as an improvement of at least one degree of lameness. 

The elbow joints were X-rayed in a slightly extended position in the mediolateral and craniocaudal beam paths without sedation (Siemens Axiom Luminos dRF). Cubarthrosis was assessed using a modified IEWG scheme, which was extended by two grades (Table 1) (www.vet-iewg.org (accessed on 20 July 2024)) [38]. To assess osteoarthritis progression, the X-ray images from the follow-up examinations were compared with those from the preliminary examination.

All treadmill examinations were carried out in the clinic’s gait analysis laboratory. 

For the computerized gait analysis, the dogs were walked on a force plate treadmill with 4 Kistler force plates (Spezialelemente Deutsche Sporthochschule Köln, Cologne, Germany) to measure the ground reaction forces, namely peak vertical force (PVF) and vertical impulse (VI). To record the kinematics, reflective markers were attached to 46 specific bone points on the forelimbs, hind limbs and spine of the dogs, which were recorded by an optical system (Vicon MX3+, Vicon Nexus Motion Systems Ltd., Oxford, UK, Quadruped Locomotion Software, in-house software of Ludwig-Maximilians Universität München). The kinematic data were analyzed to determine the range of motion (ROM) of the joints. For reporting and measurement, the present study followed the best practices for gait analysis from Conzemius et al. [39]. For all trials, the dogs were led by the same person, either the owner or a clinic employee. The optimum speed was adapted to the size and ability of each dog during the preliminary examination and used for all further gait analyses. Before all recordings, the dogs were allowed to acclimate to the treadmill until they showed a steady and natural gait pattern. Then, at least two trials of three uninterrupted minutes each were recorded. All recordings while walking and trotting were analyzed. As many steps as possible were evaluated from each trial and analyzed in graphic and numeric form. Only those steps indicating even and correct foot placement on the force plates were selected and statistically evaluated. The ground reaction forces, namely peak vertical force (PVF) and vertical impulse (VI), as well as the range of motion (ROM) were recorded. The individual therapy success of each patient was calculated with the evaluation of PVF and VI. This was defined according to Conzemius et al. (2012) as the increase in PVF and VI by ≥5% at follow-up examinations compared to the examination before treatment [40]. 

In addition to the objective and subjective lameness examinations, the owners were interviewed at each examination using three different validated questionnaires to record their assessment of their dogs’ lameness and pain. The Hudson Visual Analogue Scale (HVAS) was used to assess lameness and painfulness in dogs [41]. The HVAS Mood Score and the HVAS Movement Score were recorded, as well as the sum of the two as the HVAS Sum Score. In addition, the Liverpool Osteoarthritis in Dogs Index (LOAD) was used, in which a score is calculated from the sum of all questions [42]. The LOAD was not used from the outset of the study. It was introduced in the study protocol after the study commenced. OA-associated chronic pain was recorded using the Canine Brief Pain Inventory (CBPI) [43]. It is divided into two parts: Pain Severity Score (PSS) and Pain Interference Score (PIS). This questionnaire can also be used to record the individual patient’s therapy success [44]. Treatment success is defined as a decrease of ≥1 in PSS and ≥2 in PIS recorded during a follow-up examination in relation to the preliminary examination, which had to record values of >2 in PSS and PIS [44].

### 2.4. Follow-Up Examinations

Follow-up examinations were carried out at one, two, three, six months and, if voluntarily, at twelve months after injection. A general clinical examination, a clinical orthopedic examination and a gait analysis were carried out at all five time points. As part of each follow-up examination, the owners were asked about the side effects of the therapy. In addition, the owners completed the questionnaires described above. For this purpose, they were not informed of the results of their last questionnaires to prevent any influence. At the follow-up examinations three and six, as well as twelve months after injection, the elbow joints were X-rayed to assess osteoarthritis progression.

### 2.5. Statistical Analysis

Statistical analysis was conducted using IBM SPSS 28.0 software (IBM Deutschland GmbH, Böblingen, Germany). First, a descriptive statistical analysis of the baseline data (body weight, height at withers, age, OA score, number of injected cells) was performed. After testing for normal distribution with the Shapiro—Wilk test, the mean and standard deviation were calculated for normally distributed data. For non-normally distributed data, the median and interquartile range (IQR) were reported instead. Additionally, the range (minimum and maximum) was determined for all data.

The results of the follow-up examinations were compared with the results of the initial examination to make a statement on the development of lameness and pain after treatment. In animals treated on both sides, only the limb in which greater lameness was measured before treatment using computer-assisted gait analysis was included in the statistical analysis. The main aim of the study was therefore describing the time course of the various outcome measures (owner interview, veterinary examination with recording of the visual lameness score, computer-assisted gait analysis examination). The analysis was carried out using mixed generalized linear models. The mixed models included the dog as a grouping variable with a random intercept. Time was analyzed as a categorical variable due to different time intervals between measurements, with the time point prior to cell injection set as the reference. Covariates included in each model were weight, height at withers, age, OA score and the number of regenerative cells injected. The effects of these covariates themselves were of minor importance; rather, their corrective influence on the time course was important. The only exception to the linear model was the five-point scale of the visual lameness score. For technical reasons, an ordinal mixed model without consideration of the individual dogs had to be used as a random effect for this multinomial parameter. Patients who withdrew from the study early (after 3 months at the earliest) were also considered in the statistical analysis. To calculate the standard deviation, the results were defined in terms of the population. For all statistical analyses, *p* < 0.05 was considered significant.

## 3. Results

### 3.1. Patients

The study included 23 dogs (28 treated elbow joints: 18 dogs were treated unilaterally/5 dogs were treated bilaterally) (Table 2). Ten were female (eight of which were neutered) and thirteen were male (one of which was neutered). The following breeds were represented: seven Labrador Retrievers, five mixed breeds, three Golden Retrievers, two German Shepherds, two Flat-coated Retrievers, one Rottweiler, one Magyar Viszlar, one Black Russian Terrier and one Briard. Two dogs (both treated unilaterally) dropped out of the study after three months because the regenerative cell therapy had no effect on them and the owners wanted to have another therapy carried out. Twenty-one dogs could be followed up with for six months. In addition, six dogs could be followed up with twelve months after treatment.

### 3.2. Visual Lameness Score

The visual lameness score of the dogs decreased descriptively over time. Significant improvements were measurable at one month (*p* = 0.002), two months (*p* < 0.001) and six months (*p* = 0.002) (Table 3). On average, the patients had a score of 2.3 at the preliminary examination. Nine patients improved permanently over all follow-up examinations. Thirteen patients showed an improvement at individual follow-up examinations based on the visual lameness score. One patient showed no improvement on the lameness score.

### 3.3. Osteoarthritis Progression

The OA scores of the dogs are listed in Table 4. The mean OA score for the cohort was 3.30. Four joints showed radiographic evidence of osteoarthritis progression by one grade during the study period. The other dogs showed no radiographic evidence of osteoarthritis progression.

### 3.4. Gait Analysis

The mean PVF of all treated limbs increased descriptively over time. At the three-month follow-up, there was a significant increase compared to baseline (*p* = 0.049) (Table 3). The VI of the treated limbs increased descriptively over time and there was a significant increase six months post-injection compared to baseline (*p* = 0.037) (Table 3, Table 5). The mean values of the PVF and VI at the individual time points of the study are summarized in Table 5. During the entire study period, >33% of the dogs showed an improvement in VI and PVF of ≥5% (Table 6). The mean ROM of the treated elbow joints increased descriptively over time after therapy, with a maximum at two months post-injection (Table 3). This time point was the only one that showed a significant improvement compared to baseline (*p* = 0.019) (Table 3). The ground reaction forces could not be evaluated in three follow-up examinations (one two-month follow-up and two three-month follow-ups) due to technical problems with the treadmill.

### 3.5. Owner Questionnaires

The HVAS Sum Score decreased descriptively over time. There were significant improvements in the HVAS Sum Score at all time points post-injection compared to baseline. Individually, there were significant improvements in the HVAS Mood Score at two and three months after treatment. The HVAS Movement Score improved significantly at all time points (Table 3). The LOAD score decreased descriptively over time. Significant improvements were recorded at all time points compared to baseline (Table 3). There was a lack of questionnaire data for four patients of this questionnaire. Both the CBPI PSS and the CBPI PIS improved significantly at all time points compared to baseline. There was an improvement in both scores (CBPI PSS and PIS) after therapy in 26% of patients after one month, 35% of patients after two months, 26% of patients after three months and 30% of patients after six months (Table 7). The questionnaires of one dog for the control examination after six months were missing and could therefore not be evaluated for the statistics.

### 3.6. Long-Term Follow-Up

Six dogs could be followed up with twelve months post-injection. The mean visual lameness score improved from 2.3 pre-injection (of these six dogs) to 1.5 at twelve months post-injection, although the improvement was not significant compared to pre-injection. All questionnaires showed a significant improvement compared to baseline. The individual evaluation of the CBPI PSS and PIS revealed a therapy-related improvement in 66.7% of dogs (Table 7).

No kinetic or kinematic parameters improved significantly at this time point compared to baseline (Table 3). Additionally, 50% of the dogs showed an improvement in VI and PVF of ≥5% (Table 6).

### 3.7. Covariates

Weight had a significant influence on ROM (*p* = 0.017), HVAS Mood (*p* = 0.014), HVAS Sum (*p* = 0.044) and CBPI PIS (*p* = 0.032). The results showed that a higher weight was associated with an increasing ROM, as well as higher HVAS Mood and Sum Scores, and a higher CBPI PIS. The height at the withers of the dogs had no significant influence on the results of the investigations. Age had a significant influence on ROM (*p* = 0.013), HVAS Mood (*p* < 0.001), Visual Lameness Score (*p* < 0.001), LOAD (*p* = 0.002), HVAS Movement Score (*p* < 0.001), HVAS Sum Score (*p* < 0.001), CBPI PIS (*p* < 0.001) and CBPI PSS (*p* < 0.001). It was found that the older the animals were at the initial examination, the higher their ROM and the higher their scores on the LOAD, HVAS Movement and Sum Score, as well as CBPI PIS and PSS owner questionnaires. However, it was also found that the older the dogs were, the higher their visual lameness scores were. The OA score had a significant influence on the CBPI PIS (*p* = 0.003). Dogs with a high OA score tended to have a lower CBPI PIS. The injected regenerative cell count had a significant influence on the PFV (*p* = 0.002), HVAS Movement Score (*p* = 0.020) and HVAS Sum Score (*p* = 0.021). The higher the injected cell count, the higher the PVF values and the lower the HVAS Movement and Sum Scores.

### 3.8. Side Effects of the Therapy

No wound healing disorders or side effects of the therapy were observed in any of the dogs.

## 4. Discussion

Dogs suffering from advanced OA are often in pain and frequently show chronic lameness [9,10]. Many patients cannot be helped despite surgery or conventional medical pain therapy. The aim of this study was therefore to measure the treatment success of autologous SVF in dogs with chronic lameness due to elbow osteoarthritis. The dogs no longer responded to pain therapy. To measure treatment success, orthopedic examination, gait analysis and owner questionnaires were carried out before treatment and at one, two, three and six months after treatment. At 3-month intervals, the dogs were examined radiographically to check for osteoarthritis progression. 

The patient cohort included 23 dogs, some with bilateral elbow osteoarthritis. All dogs had developed elbow osteoarthritis due to medial compartment disease and had already been treated surgically. Two patients dropped out of the study after three months because the therapy did not result in an improvement in lameness (as measured by gait analysis and orthopedic examination). The visual lameness score improved significantly at one month, two months and six months after treatment (Table 3). The treatment success of regenerative cell therapy based on the subjective lameness examination could be demonstrated in the present study, which is comparable to the results of Black et al. (2008), who measured significant improvements in veterinary examination scores over a period of six months after treatment with autologous MSCs in dogs with elbow osteoarthritis [45]. Due to different examination methods in both studies, a direct comparison is not permissible, so the results can only show a tendency. The classification of lameness using a numerical scale, as used in the present study, did not always prove to be reliable in comparison to computer-assisted gait analysis in other studies [46,47]. This is also consistent with the results of the present study. The gait analysis and lameness examinations showed significant improvements at different follow-up examinations. The gait analysis examination showed significant improvements after three (PVF) and six months (VI), while the visual lameness examination showed an improvement in lameness after one and two months compared to baseline. A placebo effect was also demonstrated in the lameness examination by a veterinarian [40], which could also have influenced the results of the subjective visual lameness score in the present, non-blinded study.Significant improvements in gait analysis could only be measured at one time point for PVF and VI. Significance for PVF was measured at the three-month examination and for VI at the six-month examination. The gait analysis of the dogs is the only objective method in the present non-blinded study. The permanent treatment success of SVF in dogs with advanced cubarthrosis could not be objectively proven. However, on average, more force was permanently transferred to the treated limb over the study period of six months. In previous research, the gait analysis after regenerative cell therapy in dogs with elbow osteoarthritis showed contradictory results. Pavarotti et al. (2020) measured a significant improvement in PVF and VI at all follow-up examinations up to six months after the treatment of 21 dogs with osteoarthritis in various joints with autologous SVF from the fatty tissue of the lumbar region (without enzymatic preparation) [35]. In contrast, Olsen et al. (2019) measured no significant improvements in PVF and VI in the study period of six months [33]. They treated 13 dogs with elbow osteoarthritis with allogeneic MSCs from the inguinal adipose tissue, which were injected intravenously three times at two-week intervals. Similarly, Kim et al. (2019) did not measure any significant improvements after treating 28 dogs with cubarthrosis [34]. The dogs showed chronic lameness and were treated intra-articularly with allogeneic umbilical mesenchymal stem cells. Since all the studies mentioned used different procedures for the harvesting and preparation of the regenerative cells, the treatment success of regenerative cells in dogs with elbow osteoarthritis could depend on the procedure. For the treatment of dogs with allogeneic regenerative cells, the cells are often cultivated in vitro and cryopreserved, whereas in the present study, the regenerative cells were harvested and prepared immediately before injection without cryopreservation. Cryopreservation of MSCs can reduce the quality of MSC products [48]. The collection site of the adipose tissue could also influence the in vivo properties of the cell products. Several studies have demonstrated different in vitro properties of MSCs depending on the tissue source [49,50,51]. The source of adipose tissue, the preparation method, including cryopreservation if necessary, and the type of injection are therefore variables that could have a significant effect on treatment success. Further studies should compare the therapeutic effects of these different methods in dogs with elbow osteoarthritis. 

In the present study, cut-off values were used to record an improvement in lameness as a result of therapy for each dog [40]. A spontaneous improvement in PVF and VI of ≥5% in dogs with OA is rare [39] and is therefore suitable for defining a therapy-related improvement in lameness over the study period [40]. 

Based on the individual evaluations, 33% to 43% of the dogs in the present study showed a therapy-related improvement in lameness over the entire study period. Punzón et al. (2022) also used these cut-off values to measure the effect of equine umbilical stem cell xenotransplantation in dogs with hip and elbow osteoarthritis over three months [52]. In the aforementioned study, the dogs had orthopedic discomfort and showed no improvement for at least three months before enrollment in the study. After treatment with xenogeneic umbilical stem cells, 41% to 63% of the dogs showed treatment-related improvement, while 8% to 21% of the dogs improved in the placebo group of the aforementioned study. More dogs responded to xenogeneic umbilical stem cell therapy in the study by Punzón et al. (2022) than to autologous regenerative cell therapy in the present study. However, dogs with coxarthrosis were also treated in the aforementioned study, and this may have influenced the results. In more than half of the dogs in the present study, there was no measurable improvement in lameness based on gait analysis during the study period. These dogs may not have responded to the cell therapy. No improvement may have been possible in these dogs because the osteophyte growths have led to an irreversible mechanical restriction. The present study included dogs that no longer responded to conservative pain therapy and most of the dogs suffered from severe cubarthrosis. A successful therapy was therefore not only defined by a lameness examination in the present study, but also by questionnaires, which subjectively examine the effects of cell therapy on the dogs’ pain and quality of life.

However, in previous research, the results of questionnaires investigating the effect of regenerative cell therapy in dogs with OA are contradictory. While Kim et al. (2019) using the HVAS and Pavarotti et al. (2020) using the CBPI measured an improvement after treatment [34,35], the results of the LOAD and CBPI in the study by Olsen et al. (2019) showed no improvement after treatment [33]. The results of the LOAD of the present study showed a significant improvement in lameness at all time points compared to baseline. This questionnaire has a clear focus on recording the lameness of the dogs and the result is consistent with the result of the HVAS Movement Score, which also improved significantly at all follow-up examinations. A significant improvement in the HAVS Sum Score was also measured at all time points. This is comparable to the study by Kim et al. (2019), which examined the effect of umbilical stem cells in dogs with cubarthrosis [34]. In the six-month study period, the HVAS Mood and Sum Scores improved significantly at all time points, but not the HVAS Movement Score. In the present study, the HVAS Mood Score improved significantly only at the follow-up examinations after two and three months. Based on the results of the HVAS in the present study, the owners therefore saw a clear improvement in the dogs’ gait (the HVAS Movement Score improved significantly at all time points), but not always an improvement in the dogs’ quality of life and mood. The evaluation of the CBPI PSS and PIS showed a significantly improved pain situation in the dogs during the entire study period. When the dogs were individually evaluated according to the thresholds described by Brown and Farrar (2013) [44], an individual therapeutic improvement in the CBPI was observed in 26% to 35% of the dogs at all time points. However, this success rate is only comparable to the success rate of the placebo group in an allogeneic MSC study by Kim et al. (2019), where 14% to 28% of the dogs in the placebo group improved, while 46% to 54% of the dogs treated with stem cells improved during the six-month study period [34]. Since the dogs in the present study suffered from chronic pain and were no longer responding to analgesic drug therapy, this could have led to a lower success rate in the present study compared to the study by Kim et al. (2019). Based on the CBPI, treatment with autologous regenerative cells in dogs with advanced cubarthrosis may not be recommended, as it only led to therapeutic pain relief in a few patients. A direct comparison with a placebo group is missing in the present study and represents a major limitation. Due to the chronic pain and advanced stage of the disease in patients, a placebo group was not included for ethical reasons.

To determine the long-term effect of regenerative cell therapy, all patients were invited to a 12-month follow-up, but only six patients attended. The validity of the results is therefore limited. The evaluation of the LOAD and HVAS showed a significant improvement compared to the baseline examination. The gait analysis and the visual lameness score of the orthopedic examination were not significant at this time point. The mean results of PVF and VI at the 12-month check-up, compared to before the therapy, also suggest a sustained increase in the force applied to the limbs (these six patients had a mean PVF of 52.4 and VI of 17.4 at the pre-examination, compared to a PVF of 56.1 and VI of 19.1 at the 12-month check-up). The evaluation of the CBPI showed a therapeutic improvement in 67% of the patients (4/6). Additionally, 50% of the dogs showed an improvement in VI and PVF of ≥5%. In three of the six dogs, a long-term effect of the treatment with regenerative cells could be demonstrated. So far, the effect of therapy in dogs with cubarthrosis using MSCs has been investigated up to six months after injection [34,45,53]. Therefore, long-term results over 12 months with a larger number of study patients would be useful to obtain results with higher evidence.

The analysis of the covariates showed that dogs with a higher weight tended to show an improvement in ROM and the questionnaires. The results of this study contrast with previous research findings and the reason for this is unclear. A limitation of the present study was that only the weight of the dogs was recorded and not their body condition score. Pro-inflammatory cytokines are increasingly released in obesity and appear to drive the inflammatory process further in OA [54]. In addition, the injection of MSCs promoted cartilage degeneration and synovial inflammation in rats that were fed a high-fat diet and suffered from mild metabolic syndrome. The authors of the study concluded that patients with mild metabolic syndrome do not respond to MSC therapy [55]. Therefore, it should be investigated in follow-up studies whether the BCS also has an influence on the success of regenerative cell therapy in dogs. 

Dogs of an older age tended to show an improvement in ROM and the questionnaires. This result contrasts with previous research. Cartilage degenerates with age and has a reduced regenerative capacity and functionality [56]. In addition, stem cells from younger dogs are better suited for therapy, as they have a higher proliferation capacity and differentiation rate [49]. 

A higher number of injected regenerative cells was a prognostic factor associated with an increasing PVF, as well as lower HVAS Movement and Sum Scores during the study period. The higher the number of injected cells, the greater the positive effect of the therapy. Before each injection, 0.1 mL of the stromal vascular fraction was taken and tested for cell count and viability. The in vitro properties of the injected regenerative cells in the present study were not examined, and this is a further limitation of the study. 

The dogs with a higher OA score showed an increasing impairment in quality of life during the study period, as measured by the CBPI PIS. However, the study cohort had a very high mean OA score of 3.0. Only four patients had an OA score of 2 and there was one patient with an OA score of 1. Further studies should therefore investigate whether a lower OA score leads to greater therapeutic success, as dogs with a low OA score were under-represented in the present study.

The point-of-care procedure for regenerative cell harvesting and processing was carried out in the clinic. The required amount of falciform fat tissue could have been harvested from each dog. The harvesting of subcutaneous fat tissue to obtain regenerative cells would also have been possible and would not have required a laparotomy. However, it is not possible to remove sufficient subcutaneous fat tissue from all patients, as the amount of subcutaneous fat depends on the patient’s BCS [50]. But, opening the abdominal cavity is a surgical procedure that is associated with risks. These include the risk of anesthesia and the risk of infection, wound healing disorders and bleeding, so this invasive procedure cannot be recommended for every patient. In the present study, no dog experienced complications. Using the procedure, an average of 8.33 million regenerative cells were isolated and injected intra-articularly. The lowest number of cells was 3.51 million and the maximum number was 14.46 million. The number of autologous ADMSCs injected in the published studies in dogs varies from 1 million to >15 million cells [57], so a comparatively large number of regenerative cells could be obtained in the present study. The use of allogeneic stem cells is significantly less invasive for the patient and involves no additional risk of anesthesia. The “Committee for Medicinal Products for Veterinary Use” (CMPV) of the European Medicine Agency (EMA) has described screenings for infectious diseases in its guidelines for allogeneic cell therapy in dogs, as it carries the risk of transmitting diseases. In addition, antibodies against allogeneic and xenogeneic MSCs have been detected in dogs after multiple MSC transplantations, but these did not lead to side effects of the therapy [58]. But, allogeneic and xenogeneic transplantation has already been successfully carried out in dogs with cubarthrosis and has been described as safe [52,58]. An advantage of allogeneic cell therapy over autologous cell therapy is that the characteristics of the donor can be determined, ensuring that the MSCs are of the highest possible quality [49]. 

## 5. Conclusions

The results of the orthopedic examination and questionnaires showed that a single injection of autologous SVF significantly improved the lameness and quality of life of dogs with chronic lameness due to progressive, therapy-resistant elbow osteoarthritis over six months. This was not confirmed by the objective gait analysis, as no permanent significant improvement in lameness was measured. However, some of the dogs showed a therapeutically induced improvement in lameness and pain over six months. Further studies should investigate the effect of SVF in dogs with advanced OA. 

## Figures and Tables

**Table 1 animals-14-02803-t001:** Modified, extended IEWG scheme.

OA Score	Osteophytes in mm
0	0
1	0–2
2	2–5
3	5–7
4	7–9
5	>9

**Table 2 animals-14-02803-t002:** Morphometric data of the 23 dogs.

	Minimum	Maximum	Mean ± Standard Deviation
**Weight**	18.0 kg	60.0 kg	34.5 ± 10.6 kg
**Height at withers**	45.0 cm	70.0 cm	59.7 ± 6.6 cm
**OA Score**	1	5	3.3 ± 1.1
**Number of injected regenerative cells**	3.5 Mio	14.5 Mio	8.2 ± 3.0 Mio
	**Min**	**Max**	**Median interquartile range**
**Age**	0.8 years	12.0 years	8.3 2.8–10.0 years

**Table 3 animals-14-02803-t003:** Statistical analysis of the owner questionnaires: Liverpool Osteoarthritis in Dogs Index (LOAD), Canine Brief Pain Inventory (CBPI) and Hudson Visual Analogue Scale (HVAS). Visual lameness score and gait analysis. Peak vertical force (PVF), vertical impulse (VI) and range of motion (ROM). Significance (*p* < 0.05) is marked with *.

	LOAD	CBPI PSS	CBPI PIS	HVAS Mood Score	HVAS Movement Score	HVAS Sum Score	Visual Lameness Score	PVF	VI	ROM
1 month	*p* < 0.001 *	*p* = 0.002 *	*p* < 0.001 *	*p* = 0.105	*p* < 0.001 *	*p* = 0.001 *	*p* = 0.002 *	*p* = 0.080	*p* = 0.157	*p* = 0.453
2 months	*p* < 0.001 *	*p* = 0.0002 *	*p* < 0.001 *	*p* = 0.017 *	*p* < 0.001 *	*p* < 0.001 *	*p* < 0.001 *	*p* = 0.100	*p* = 0.108	*p* = 0.019 *
3 months	*p* < 0.001 *	*p* = 0.017 *	*p* = 0.003 *	*p* = 0.046 *	*p* < 0.001 *	*p* = 0.002 *	*p* = 0.070	*p* = 0.049 *	*p* = 0.089	*p* = 0.554
6 months	*p* = 0.004 *	*p* = 0.020 *	*p* = 0.005 *	*p* = 0.064	*p* = 0.007 *	*p* = 0.010 *	*p* = 0.002 *	*p* = 0.100	*p* = 0.037 *	*p* = 0.242
12 months	*p* = 0.006 *	*p* = 0.020 *	*p* = 0.009 *	*p* = 0.033 *	*p* = 0.015 *	*p* = 0.011 *	*p* = 0.366	*p* = 0.090	*p* = 0.077	*p* = 0.863

**Table 4 animals-14-02803-t004:** Modified and expanded IEWG score of the joints that were statistically analyzed.

OA Score	Osteophytes in mm	Number of Joints
0	0	0
1	0.1–2	1
2	2.1 bis 5	4
3	5.1 bis 7	7
4	7.1 bis 9	9
5	>9	2
Total		23

**Table 5 animals-14-02803-t005:** Results of PVF and VI at all examinations. Summarized as mean ± standard deviation.

	Baseline	1 Month	2 Months	3 Months	6 Months	12 Months
**Number of dogs examined**	23	23	22	21	21	6
**PVF**	52.8 ± 6.8	54.9 ± 5.5	55.0 ± 8.5	55.4 ± 7.4	54.7 ± 6.6	56.1 ± 4.6
**VI**	17.2 ± 2.2	18.0 ± 2.3	18.3 ± 3.1	18.3 ± 2.2	18.5 ± 2.8	19.1 ± 1.7

**Table 6 animals-14-02803-t006:** Number of dogs in % and absolute terms that showed a clinical improvement in PVF and VI of ≥5% at the individual follow-up examinations in relation to the baseline examination.

(Number of Dogs Examined)	1 Month (23)	2 Months (22)	3 Months (21)	6 Months (21)	12 Months (6)
**PVF and VI ≥ 5%**	8 (35%)	9 (41%)	7 (33%)	9 (43%)	3 (50%)

**Table 7 animals-14-02803-t007:** Number of dogs in % and absolute terms that showed a clinical improvement based on the CBPI at the individual follow-up examinations in relation to the baseline examination.

Time Point (Number of Dogs)	1 Month (23)	2 Months (23)	3 Months (23)	6 Months (20)	12 Months (6)
**Number of dogs (in%)**	6 (26%)	8 (35%)	6 (26%)	6 (30%)	4 (67%)

## Data Availability

The data that support the findings of this study are available from the corresponding author.

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
