# Peer review of "Efficacy of a Single Injection of Stromal Vascular Fraction in Dogs with Elbow Osteoarthritis: A Clinical Prospective Study"

_animals, 2024, doi:10.3390/ani14192803_

Round 1

Reviewer 1 Report

Comments and Suggestions for Authors

This is overall a well written manuscript that tries to evaluate the use of autologous SVF in dogs to treat pain associated with OA. There is a need to improve our treatment options for dogs with osteoarthritis and investigations of regenerative treatment are important. The major limitation of this study is the lack of a control group. In addition, this reviewer would encourage the authors to be a bit more cautious when interpreting their results as the objective evidence for improvement was not strong. 

Line 112: The authors state that a modified IEWG score of 2 was a prerequisite but on line 281 (Table 3) there is 1 joint listed with a score of 1. Should this joint not be excluded from statistical analysis?

Line 165-166: How was range of motion assessed? Goniometry measurement would be standard. If not used, this author questions the validity of very subjectively flexing and extending the joint once a month and determining if the degree has changed over time. If goniometry was used, please report the actual change in degrees. If not used, I recommend removing this from the study reporting. 

Line 187-188: With regards to gait analysis, it is standard to control the speed and acceleration and typically record 5 valid trials to decrease variability. The protocol used in the current study related to the gait analysis should be better described,  so the reader can understand what the protocol was. Further, please discuss why the speed and acceleration were not standardized across the dogs, as is generally done. This may very well be a limitation to the study. 

Line 192-194: I am not certain this is correctly interpreted. If I understand your statement correctly, you are saying that an increase in VI and PVF of less than 5% at follow up compared to pretreatment indicated treatment success. In reference 39, lameness was improved if there was a > or = 5% change in VI or PVF. Is this what the authors meant to say?

Results: 

The data presented in table 1 is not described in the statistical analysis section. Are the values (height, weight, age, OA score and injected number of cells) all normally distributed? If so, how was this assessed? Please add this information to the statistical analysis section. 

Table 5 is a little misleading as at 12 months, it is possible that the 17 dogs not gaited perhaps did not have a >=5% improvement. It is possible that owners with no obvious improvement in their dogs elected not to come back.

Discussion:

This reviewer recommends a short discussion on interpreting subjective data versus objective data. In this study, there was minimal evidence that the treatment worked based on the objective gait data. We know from several studies that the caregiver effect can influence the interpretation of success, rendering outcome measures, such as lameness grade, less valuable. 

Line 367-368: Where these dogs excluded or did they drop out (as described in the results section)?  It would make no sense to exclude them and lack of improvement would be an important finding and not a reason to exclude. 

Line 379-382: This reviewer would be cautious with overstating the significant improvement here as it only happened at 1 time point for PVF and 1 time point for VI during the study period. This would be an important discussion point as this is the only objective outcome measure in this non-blinded study. 

Line 487-488: There is amble evidence that increase in weight/BCS worsens clinical signs of OA - this is something the authors can discuss as the results in this study are quite contradictory to this. I do however question the strength of the ROM data collected as it is not clear if it was objective or subjective. If subjective, I do recommend that it is removed. 

Line 504-505: In table 3, the authors report 1 point with the score of 1. Please resolve this discrepancy.

Line 532-533: These 2 sentences are a bit odd here. Either remove or incorporate a bit more into the text. 

Comments on the Quality of English Language

Overall, the English language used in the manuscript is good and only requires a small amount of editing to make it ready for potential publication. 

Author Response

Thank you very much for taking the time to review this manuscript and for your constructive feedback. Please find the detailed responses below and the corresponding corrections highlighted in the re-submitted files.

Line 112: The authors state that a modified IEWG score of 2 was a prerequisite but on line 281 (Table 3) there is 1 joint listed with a score of 1. Should this joint not be excluded from statistical analysis?

Thank you, we agree. It was a typing error and we corrected the sentence.

Line 120: A modified OA score according to the protocol of the International Elbow Working Group (IEWG score) of 1 was a prerequisite for inclusion in the study.

Line 165-166: How was range of motion assessed? Goniometry measurement would be standard. If not used, this author questions the validity of very subjectively flexing and extending the joint once a month and determining if the degree has changed over time. If goniometry was used, please report the actual change in degrees. If not used, I recommend removing this from the study reporting. 

Thank you for pointing this out. As it may not be clear to the reader, we have reworded the sentence. During the orthopaedic examination, only restricted mobility was tested. The range of motion was recorded during the computerized gait analysis, which has now been described in more detail.

Line 171-173: During the orthopaedic examination, the elbow joints were palpated to detect any increased joint filling. The joints were passively manipulated so that a restricted mobility as well as painful defensive reactions or crepitation could be recorded.

Line 189-297: All treadmill examinations were carried out in the clinic’s gait analysis laboratory. For the computerized gait analysis, the dogs were walked on a force plate treadmill with 4 Kistler force plates (Spezialelemente Deutsche Sporthochschule Köln, Cologne, Germany) to measure the ground reaction forces Peak Vertical Force (PVF) and Vertical Impulse (VI). To record the kinematics, reflective markers were attached to 46 specific bone points on the forelimbs, hind limbs and spine of the dogs, which were recorded by an optical system (Vicon Nexus Motion Systems Ltd, Oxford, United Kingdom, Quadruped Locomotion Software). The kinematic data was analyzed to determine the range of motion (ROM) of the joints.   

Line 187-188: With regards to gait analysis, it is standard to control the speed and acceleration and typically record 5 valid trials to decrease variability. The protocol used in the current study related to the gait analysis should be better described, so the reader can understand what the protocol was. Further, please discuss why the speed and acceleration were not standardized across the dogs, as is generally done. This may very well be a limitation to the study. 

Thank you, we have described the gait analysis in more detail. For reporting and measurement, we have followed the best practices of Conzemius et al. Since the present study population was a heterogeneous group, we opted for a preferred speed for each dog in order to keep the number of recordings low.

Lines 197-206: For reporting and measurement, we have followed the best practices of Conzemius et al. [1]. For all trials, the dogs were led by the same person, either the owner or a clinic employee. The optimum speed was adapted to the size and ability of each dog during the preliminary examination and used for all further gait analyses. Before all recordings, the dogs were allowed to acclimate to the treadmill until they showed a steady and natural gait pattern. Then, at least two trials of three uninterrupted minutes each were recorded. All recordings while walking and trotting were analyzed. As many steps as possible were evaluated from each trial and analyzed in graphic and numeric form. Only those steps indicating even and correct foot placement on the force plates were selected and statistically evaluated.

Line 192-194: I am not certain this is correctly interpreted. If I understand your statement correctly, you are saying that an increase in VI and PVF of less than 5% at follow up compared to pretreatment indicated treatment success. In reference 39, lameness was improved if there was a > or = 5% change in VI or PVF. Is this what the authors meant to say?

Thank you very much. Please excuse, it was a typing error. We have changed it to:

Line 209-211: This was defined according to Conzemius et al. (2012) as the increase in PVF and VI by ≥5 % at follow-up examinations compared to the examination before treatment

Results: 

The data presented in table 1 is not described in the statistical analysis section. Are the values (height, weight, age, OA score and injected number of cells) all normally distributed? If so, how was this assessed? Please add this information to the statistical analysis section. 

Thank you for pointing this out. We have, accordingly, revised the statistical analysis section and the table 2 (line 277-278):

Line 237-242: First, a descriptive statistical analysis of the baseline data (body weight, height at withers, age, OA-score, number of injected cells) was performed. After testing for normal distribution with the Shapiro-Wilk test, the mean and standard deviation were calculated for normally distributed data. For non-normally distributed data, the median and interquartile range (IQR) were reported instead. Additionally, the range (minimum and maximum) was determined for all data.

Table 5 is a little misleading as at 12 months, it is possible that the 17 dogs not gaited perhaps did not have a >=5% improvement. It is possible that owners with no obvious improvement in their dogs elected not to come back.

We appreciate you bringing this to our attention. To make it clearer, we have added the word ´´examined´´in the table 5, so that the reader is aware that only 6 dogs appeared for the follow-up examination after 12 months (320-324).

We agree, that the interpretation of the 12-month examination should be with caution and have therefore described it separately in the results section (line 348-357). Consequently, we have provided a detailed description of the limitation in the discussion (lines 499-513).

Discussion:

This reviewer recommends a short discussion on interpreting subjective data versus objective data. In this study, there was minimal evidence that the treatment worked based on the objective gait data. We know from several studies that the caregiver effect can influence the interpretation of success, rendering outcome measures, such as lameness grade, less valuable. 

 Thank you, we agree with you and have emphasized this more clearly in the discussion.

Line 407-409: A placebo effect was also demonstrated for the lameness examination by a veterinarian [2], which could also have influenced the results of the subjective visual lameness score in the present, non-blinded study.

Line 413-415: The gait analysis of the dogs is the only objective method in the present non-blinded study. A permanent treatment success of SVF in dogs with advanced cubarthrosis could not be objectively proven.

Line 463-466: A successful therapy was therefore not only defined by a lameness examination in the present study, but also by questionnaires, which subjectively examine the effects of cell therapy on the dogs' pain and quality of life.

Line 567-574: The results of the orthopaedic examination and questionnaires showed that a single injection of autologous SVF significantly improved the lameness and quality of life of dogs with chronic lameness due to progressive, therapy-resistant elbow osteoarthritis over six months. This was not confirmed by the objective gait analysis, as no permanent significant improvement in lameness was measured. However, some of the dogs showed a therapeutically induced improvement in lameness and pain over six months. Further studies should investigate the effect of SVF in dogs with advanced OA.

Line 367-368: Where these dogs excluded or did they drop out (as described in the results section)?  It would make no sense to exclude them and lack of improvement would be an important finding and not a reason to exclude. 

Thank you for pointing this out. We have revised the sentence.

Line 391-392: Two patients dropped out of the study after three months because the therapy did not result in an improvement in lameness (as measured by gait analysis and orthopaedic examination).

Line 379-382: This reviewer would be cautious with overstating the significant improvement here as it only happened at 1 time point for PVF and 1 time point for VI during the study period. This would be an important discussion point as this is the only objective outcome measure in this non-blinded study. 

Thank you, we agree with you and have added it to the following paragraph:

Line 411-415: Significant improvements in gait analysis could only be measured at one time point for PVF and VI. Significance for PVF was measured at the three-month and for VI for the six-month examination. The gait analysis of the dogs is the only objective method in the present non-blinded study. No permanent treatment success of SVF in dogs with advanced cubarthrosis could be objectively proven.

Line 487-488: There is amble evidence that increase in weight/BCS worsens clinical signs of OA - this is something the authors can discuss as the results in this study are quite contradictory to this. I do however question the strength of the ROM data collected as it is not clear if it was objective or subjective. If subjective, I do recommend that it is removed. 

Thanks for pointing this out. I have expanded the discussion with additional aspects. The ROM was kept as it was part of the objective data collection of the gait analysis.

Line 514-524: The analysis of the covariates showed that dogs with a higher weight tended to have an improvement in ROM and the questionnaires. The result of this study contrasts with previous research findings and the reason for this is unclear. A limitation of the present study was that only the weight of the dogs was recorded and not their body condition score. Proinflammatory cytokines are increasingly released in obesity and appear to drive the inflammatory process further in OA [3]. In addition, the injection of MSCs promoted cartilage degeneration and synovial inflammation in rats that were fed a high-fat diet and suffered from mild metabolic syndrome. The authors of the study concluded that patients with mild metabolic syndrome do not respond to MSC therapy [4]. Therefore, it should be investigated in follow-up studies whether the BCS also has an influence on the success of regenerative cell therapy in dogs.

Line 525-529Dogs with an older age tended to have an improvement in ROM and the questionnaires. This result contrasts with previous research. Cartilage degenerates with age and has a reduced regenerative capacity and functionality [5]. In addition, stem cells from younger dogs are better suited for therapy, as they have a higher proliferation capacity and differentiation rate [6].

Line 504-505: In table 3, the authors report 1 point with the score of 1. Please resolve this discrepancy.

Thank you for pointing this out. I have resolved it:

Line 543-544: Only four patients had an OA score of 2 and there was one patient with an OA score of 1.

Line 532-533: These 2 sentences are a bit odd here. Either remove or incorporate a bit more into the text. 

Thank you, we concur and have removed the sentences.

Reviewer 2 Report

Comments and Suggestions for Authors

Materials & Methods: 

  • Lines 104-5: Were injections of any other substance included? For instance, some platelet products and hydrogels may have positive effects lasting longer than 3 months, this should be clarified.  

  • Line 112: Please write out what IEWG stands for if this is the first appearance in the manuscript.  

  • Lines 129-130: Rescue analgesics utilized should ideally be described. The authors should also discuss if nutraceuticals and diets were kept consistent throughout the study.  

  • Line 193: Should this be an increase in PVF and VI by >/=5%, (rather than less than)? 

Results: 

  • Lines 253-254 (Table 1): change the commas to decimal points 

  • Lines 296-297 (Table 4): change the commas to decimal points 

  • Lines 301-303 (Table 5): The authors should consistently round all of the percentages to the same decimal place. If rounding to the 10th or 100th of a percent, please change the commas to decimal points. It is currently very confusing as written. 

  • Lines 320-322 (Table 6): change the commas to decimal points 

Discussion: 

  • Overall - The discussion is lengthy but appropriate; the authors could consider dividing the discussion into sub-sections (similar to the results section) to make it more reader-friendly.  

  • Lines 535-539: This conclusion is an overstatement given this study was non-blinded and uncontrolled, and osteoarthritis is known to wax and wane. The authors should consider concluding that based on these results, autologous SVF may yield improvements in some dogs with osteoarthritis, however further controlled and blinded studies are indicated.

Author Response

Thank you very much for taking the time to review this manuscript and for your constructive feedback. Please find the detailed responses below and the corresponding corrections highlighted in the re-submitted files.

  • Lines 104-5: Were injections of any other substance included? For instance, some platelet products and hydrogels may have positive effects lasting longer than 3 months, this should be clarified.

Thank you for pointing this out. Therefore we added the information to the sentences :

Lines 108-109: Study patients who underwent bilateral arthroscopy for medial compartment disease and had bilateral OA with clinical relevance were included in the study for bilateral treatment with SVF. Unilaterally operated dogs that had only unilateral clinically relevant OA were treated unilaterally with SVF.

  • Line 112: Please write out what IEWG stands for if this is the first appearance in the manuscript.

Thank you, we agree and corrected the sentence.

Lines 119-120: A modified OA score according to the protocol of the International Elbow Working Group (IEWG score) of 1 was a prerequisite for inclusion in the study.

  • Lines 129-130: Rescue analgesics utilized should ideally be described. The authors should also discuss if nutraceuticals and diets were kept consistent throughout the study.

We agree and have supplemented the sentence:

Lines 136-138: No patient received permanent analgesic therapy during the study period; emergency administration of an analgesic (NSAID or metamizol) was permitted in the event of worsening lameness or clear signs of pain.

  • Line 193: Should this be an increase in PVF and VI by >/=5%, (rather than less than)?

Thank you very much. Please excuse, it was a typing error. We have changed it to:

Lines 209-210: This was defined according to Conzemius et al. (2012) as the increase in PVF and VI by ≥5 % at follow-up examinations compared to the examination before treatment

Results: 

  • Lines 253-254 (Table 1): change the commas to decimal points

Thank you, we changed all commas to decimal points (lines 277 to 278)

  • Lines 296-297 (Table 4): change the commas to decimal points

Thank you, we changed all commas to decimal points (lines 320-322)

  • Lines 301-303 (Table 5): The authors should consistently round all of the percentages to the same decimal place. If rounding to the 10th or 100th of a percent, please change the commas to decimal points. It is currently very confusing as written.

Thank you, we rounded all the percentages and changed all commas to decimal points (lines 310 to 311).

  • Lines 320-322 (Table 6): change the commas to decimal points

Thank you, we changed all commas to decimal points (lines 327 to 328)

Discussion: 

  • Overall - The discussion is lengthy but appropriate; the authors could consider dividing the discussion into sub-sections (similar to the results section) to make it more reader-friendly.

We acknowledge your viewpoint and appreciate your input. However, we would prefer to maintain the discussion in its current format, without the inclusion of subsections.

  • Lines 535-539: This conclusion is an overstatement given this study was non-blinded and uncontrolled, and osteoarthritis is known to wax and wane. The authors should consider concluding that based on these results, autologous SVF may yield improvements in some dogs with osteoarthritis, however further controlled and blinded studies are indicated.

Thank you. We have, accordingly, revised the conclusion to emphasize this point.

Lines 572-578: The results of the orthopaedic examination and questionnaires showed that a single injection of autologous SVF significantly improved the lameness and quality of life of dogs with chronic lameness due to progressive, therapy-resistant elbow osteoarthritis over six months. This was not confirmed by the objective gait analysis, as no permanent significant improvement in lameness was measured. However, some of the dogs showed a therapeutically induced improvement in lameness and pain over six months. Further studies should investigate the effect of SVF in dogs with advanced OA.

Reviewer 3 Report

Comments and Suggestions for Authors

Thank you for your manuscript. Regenerative therapy for osteoarthritis is of great interest and yet is a challenge given all the variability in source of MSC, preparation methods, etc. and individual patient responsiveness of which we know very little about intrinsic factors which may influence patient response to regenerative therapies aside from OA severity which has been demonstrated to result in poorer outcomes with a vast majority of treatment options. I have provided some specific comments for areas of the manuscript I would suggest editing to improve the readibility and quality of the paper: 

Title: I would recommend editing the title from “regenerative cell therapy” to say single injection of stromal vascular fraction. This will help the reader to immediately identify what will be discussed in this paper and help compare to other studies that may use different types of regenerative medicine or multiple injection protocols. 

Line 30: Typo “impuls” should be “impulse” 

I would appreciate more detail on duration of lameness/pain you are using to determine these dogs had chronic pain despite conventional treatment as well as the duration of treatment these dogs had received prior to enrollment. Were all dogs receiving consistent and appropriate doses of NSAID prior to enrollment? Were any adjunctive analgesics or conservative management techniques utilized in these patients (nutraceuticals, physiotherapy, etc.). If any of the patients had received prior intra-articular injections, this should also be described. 

Line 112: Please type out International Elbow Working Group here in case readers aren’t familiar with the IEWG abbreviation. 

Also Line 112: I am confused that you state that a modified IEWG score of 2 was a prerequisite for inclusion in the study, but there was a dog included who scored 1. 

Line 193: Should this say greater than or equal to 5% instead of less than/equal to? 

Line 293-294 and Line 317: Using “control” here is confusing as it makes me think you are referring to dogs in a control group vs the treated group which isn’t correct since you did not use a control group. Please reword. Also, please state why ground reaction forces could not be evaluated in these dogs at these time points. 

Please provide a table with the HVAS, LOAD, and CBPI raw data at each of the time points. 

Line 416-417: Incomplete sentence, please edit 

Line 425-428: These sentences are confusing and seem contradictory that the dogs in this other study did not show improvement for at least 3 months but then the next sentence says the percent of dogs that did show treatment related improvement, please edit. 

Line 446: Typo “consistend” should be “consistent” 

Line 532-533: I suggest removing these sentences here, they are not needed. 

Line 536: Please specify single injection stromal vascular fraction instead of just regenerative cell therapy here and include duration that efficacy was assessed/seen to make the conclusion statement stronger/more complete.

Author Response

Thank you very much for taking the time to review this manuscript and for your constructive feedback and specific comments. Please find the detailed responses below and the corresponding corrections highlighted in the re-submitted files.

Title: I would recommend editing the title from “regenerative cell therapy” to say single injection of stromal vascular fraction. This will help the reader to immediately identify what will be discussed in this paper and help compare to other studies that may use different types of regenerative medicine or multiple injection protocols. 

We acknowledge your viewpoint and appreciate your input. We have changed the title to: Efficacy of a Single Injection of Stromal Vascular Fraction in Dogs with Elbow Osteoarthritis: a Clinical Prospective Study

Line 30: Typo “impuls” should be “impulse” 

Thank you. We corrected this typing error (line 30).

I would appreciate more detail on duration of lameness/pain you are using to determine these dogs had chronic pain despite conventional treatment as well as the duration of treatment these dogs had received prior to enrollment. Were all dogs receiving consistent and appropriate doses of NSAID prior to enrollment? Were any adjunctive analgesics or conservative management techniques utilized in these patients (nutraceuticals, physiotherapy, etc.). If any of the patients had received prior intra-articular injections, this should also be described. 

Thank you for pointing this out. We have supplemented the information as follows:

Line 100-107: To be included in the study, the dogs had to exhibit lameness for at least three months that no longer showed any improvement despite adequate pain therapy. A specific treatment protocol was not prescribed prior to inclusion in the therapy, as the choice of painkiller was based on the dogs' tolerance. There was no requirement for the dogs to have received neutraceuticals or physiotherapy prior to inclusion in the study; only that the dogs receiving these therapies prior to inclusion in the study had to continue them constantly during the study in order to rule out any influence on the study results.

Line 112: Please type out International Elbow Working Group here in case readers aren’t familiar with the IEWG abbreviation

Thank you, we agree and corrected the sentence.

Line 119: A modified OA score according to the protocol of the International Elbow Working Group (IEWG score) of 1 was a prerequisite for inclusion in the study.

Also Line 112: I am confused that you state that a modified IEWG score of 2 was a prerequisite for inclusion in the study, but there was a dog included who scored 1.

We would like to thank you for bringing this typing error to our attention. We have corrected it (line 112):

Thank you for pointing this out. Excuse, it was a typing error.

Line 119: A modified OA score according to the protocol of the International Elbow Working Group (IEWG score) of 1 was a prerequisite for inclusion in the study.

Line 193: Should this say greater than or equal to 5% instead of less than/equal to? 

Thank you, we have corrected the sentence.

Line 210: This was defined according to Conzemius et al. (2012) as the increase in PVF and VI by ≥5 % at follow-up examinations compared to the examination before treatment

Line 293-294 and Line 317: Using “control” here is confusing as it makes me think you are referring to dogs in a control group vs the treated group which isn’t correct since you did not use a control group. Please reword. Also, please state why ground reaction forces could not be evaluated in these dogs at these time points. 

Thank you, we have accordingly revised the sentence:

Line 316-318: The ground reaction forces could not be evaluated in three follow-up examinations (one two-month follow-up and two three-month follow-ups) due to technical problems with the treadmill.

Please provide a table with the HVAS, LOAD, and CBPI raw data at each of the time points.

Thank you. We have provided the table as supplementary material.

Line 416-417: Incomplete sentence, please edit 

Thank you. We agree and in order to enhance the coherence of the paragraph for the reader, we have shortened it.

Line 441-443: In the present study, cut-off values were used to record an improvement in lameness as a result of therapy for each dog [2]. A spontaneous improvement in PVF and VI of ≥5 % in dogs with OA is rare [1] and is therefore suitable for defining a therapy-related improvement in lameness over the study period [2].

Line 425-428: These sentences are confusing and seem contradictory that the dogs in this other study did not show improvement for at least 3 months but then the next sentence says the percent of dogs that did show treatment related improvement, please edit. 

Thank you, we have accordingly revised the sentence.

Line 451-452: In the aforementioned study the dogs had orthopaedic discomfort and showed no improvement for at least three months before enrollment in the study.

Line 446: Typo “consistend” should be “consistent” 

Thank you for bringing this to our attention. We have amended the spelling (line 474).

Line 532-533: I suggest removing these sentences here, they are not needed. 

Thank you for your suggestion, we have removed the sentences.

Line 536: Please specify single injection stromal vascular fraction instead of just regenerative cell therapy here and include duration that efficacy was assessed/seen to make the conclusion statement stronger/more complete.

Thank you for pointing this out. We have revised the conclusion.

Lines 568-574: The results of the orthopaedic examination and questionnaires showed that a single injection of autologous SVF significantly improved the lameness and quality of life of dogs with chronic lameness due to progressive, therapy-resistant elbow osteoarthritis over six months. This was not confirmed by the objective gait analysis, as no permanent significant improvement in lameness was measured. However, some of the dogs showed a therapeutically induced improvement in lameness and pain over six months. Further studies should investigate the effect of SVF in dogs with advanced OA.

Round 2

Reviewer 1 Report

Comments and Suggestions for Authors

Thank you for the revisions and development of this manuscript. I believe that the discussion of the results are thoughtful and accurate without over-interpreting the results. 

Comments on the Quality of English Language

appropriate